# Therapeutic Significance of microRNA-Mediated Regulation of PARP-1 in SARS-CoV-2 Infection

**DOI:** 10.3390/ncrna7040060

**Published:** 2021-09-22

**Authors:** Sabyasachi Dash, Chandravanu Dash, Jui Pandhare

**Affiliations:** 1Weill Cornell Medicine, Department of Pathology and Laboratory Medicine, Cornell University, New York, NY 10065, USA; 2Center for AIDS Health Disparities Research, Meharry Medical College, Nashville, TN 37208, USA; cdash@mmc.edu (C.D.); jpandhare@mmc.edu (J.P.); 3School of Graduate Studies and Research, Meharry Medical College, Nashville, TN 37208, USA; 4Department of Biochemistry, Cancer Biology, Pharmacology and Neuroscience, Meharry Medical College, Nashville, TN 37208, USA; 5Department of Microbiology, Immunology and Physiology, Meharry Medical College, Nashville, TN 37208, USA

**Keywords:** COVID-19, PARP-1, miRNA, therapeutics, drug repurposing, neuropathology, RNA viruses, SARS-CoV-2

## Abstract

The COVID-19 pandemic caused by the novel coronavirus SARS-CoV-2 (2019-nCoV) has devastated global healthcare and economies. Despite the stabilization of infectivity rates in some developed nations, several countries are still under the grip of the pathogenic viral mutants that are causing a significant increase in infections and hospitalization. Given this urgency, targeting of key host factors regulating SARS-CoV-2 life cycle is postulated as a novel strategy to counter the virus and its associated pathological outcomes. In this regard, Poly (ADP)-ribose polymerase-1 (PARP-1) is being increasingly recognized as a possible target. PARP-1 is well studied in human diseases such as cancer, central nervous system (CNS) disorders and pathology of RNA viruses. Emerging evidence indicates that regulation of PARP-1 by non-coding RNAs such as microRNAs is integral to cell survival, redox balance, DNA damage response, energy homeostasis, and several other cellular processes. In this short perspective, we summarize the recent findings on the microRNA/PARP-1 axis and its therapeutic potential for COVID-19 pathologies.

## 1. Introduction

The rapidly evolving coronavirus disease (COVID-19), caused by the severe acute respiratory syndrome coronavirus 2 (SARS-CoV-2), presents an urgent and unmet clinical need for novel therapeutics [1,2]. This effort requires a clear understanding of the role of host cell factors that aid in the viral life cycle. This knowledge will pave the way for efficient drug repurposing and novel therapeutic strategies. One such host factor that has caught considerable attention is Poly (ADP)-ribose polymerase-1 (PARP-1) [3,4,5,6]. PARP-1 is a multi-domain enzyme that utilizes cellular NAD+ to catalyze the synthesis of poly (ADP) ribose (PAR) residues following its transfer onto target proteins by a mechanism known as PARylation [7,8,9]. PARP-1 belongs to the PARP/ARTD family of enzymes that constitutes 17 isoforms in humans (PARP-1–PARP-17) [10]. It is a highly conserved enzyme with a molecular mass of 116 kDa [7] that predominantly localizes in the nucleus. PARP-1 multi-domain unit comprises the amino (*N*) terminal DNA binding domain (DBD), a central auto-modification domain (AMD), Zn binding domain (FIII/Zn3), and a carboxyl (C) terminal catalytic domain [11], as illustrated in Figure 1 [12,13,14]. Among the different isoforms, PARP-1 is predominantly responsible for 85–90% of cellular PARP activity; PARP-2 is responsible for 10–15%; and the remainder of the enzymes contribute to the remainder of PARP activity [8,15]. Like the PARP-1 function, recent studies have highlighted the role of PARP-2 in response to DNA damage, genome maintenance, metabolic regulation, and oxidative stress [16,17]. Alteration in PARP-1 activity has been reported in several diseases including viral pathogenesis [9,18]. For instance, interferon-stimulated genes (ISGs), which are members of the PARP superfamily, have been implicated in the restriction of viral replication. However, the exact mechanisms underlying the antiviral response of the PARP superfamily of proteins are not clearly understood [19,20]. This is because several viruses harbor the machinery to modulate PARP function, implying that PARP-mediated signaling can be both anti- and pro-viral in a context-dependent manner [20,21]. However, with regards to coronavirus species, a recent study reported that SARS-CoV-2 infection strikingly induced activity of PARPs, along with expression of genes encoding enzymes for salvage NAD synthesis from nicotinamide and nicotinamide riboside, while simultaneously downregulating other NAD utilizing biosynthetic pathways [4,10,13,22].

PARP-1 is a multi-functional enzyme, which depends on specific interactions with DNA, nucleosomes, or chromatin-associated proteins [11,22,23,24]. PARP-1-mediated PARylation of proteins is a key post-translational modification that leads to a series of molecular signaling cascades involving ATP and co-factor NAD+ [25]. In addition, PARP-1 undergoes auto-modification, i.e., auto-PARylation, marked by the addition of ADP-ribose chains ranging from 20 to more than 200 units in length, that inhibit its DNA binding ability and regulate its catalytic activity [7]. Thus, PARylation is a critical determinant for direct (autocrine) and indirect (paracrine) molecular signaling that is dependent on PARP-1 function. In this perspective, we summarize and discuss two important questions pertaining to the miRNA-mediated regulation of PARP-1 in the context of COVID-19 pathogenesis. First, how relevant is PARP-1’s activity across cell/tissue types in SARS-CoV-2 pathogenesis and second, can miRNA-mediated regulation of PARP-1 be exploited as a therapeutic target to prevent SARS-CoV-2 infection to mitigate COVID-19 pathological outcomes?

## 2. PARP-1 in Viral Pathology

Emerging evidence highlights the role of the PARP family of proteins, and specifically PARP-1, as a potential therapeutic target for COVID-19 [3,4,5,6]. Various small molecule inhibitors targeting PARP-1′s enzymatic activity are available and have been extensively studied. Currently, these inhibitors are under investigation in clinical trials for several cancer types and associated disorders [26,27]. *Unfortunately, only a limited number of studies have discussed the repurposing of PARP1 inhibitors in other disease models (e.g., inflammatory or viral diseases)*. Given the ability of PARP proteins to bind nucleic acids, several members of the PARP family have been shown to impact host–virus interactions. In particular, these proteins affect a number of steps of the viral life cycle (integration, recombination, and transcription) suggesting potential pro- and antiviral properties of PARP proteins [20,28,29]. Although these findings are primarily reported for other RNA viruses, such as influenza and porcine reproductive and respiratory syndrome viruses (PRSSV), the effect of PARP-1 and other PARP proteins on coronavirus species cannot be ignored. In general, studies have highlighted that PARP-1 inhibitors may be useful therapeutic targets for viral infections [5,6]. However, the drug doses optimized for treating cancers may be insufficient to treat viral infection and pathogenesis. Hence, there is an urgent need for research to understand the impact of drugs targeting PARP1 on viral pathogenesis, specifically investigation of the direct role of PARP-1 and SARS-CoV-2 life cycle and pathology.

As illustrated in Figure 2, in SARS-CoV-2 infection, both viral proteins and virions in circulation bind to cell surface receptors (e.g., ACE2 and TLRs) and activate immune response cascades (e.g., pro-inflammatory cytokines such as IL6), in addition to increasing mitochondrial reactive oxygen species (ROS) and impairing redox balance, followed by PARP-1-induced PARylation of downstream targets, which results in depletion of intracellular ATP levels causing cellular apoptosis and tissue injury [30,31,32]. Viruses require an intracellular ATP pool to maintain their life cycle in the host cell [33,34]. Viral replication is dependent on nicotinamide adenine dinucleotide (NAD) coenzymes and redox factors such as NAD+, NADH, NADP+, and NADPH, which are central to cellular metabolism [35]. Primarily, these coenzymes regulate electron (e^−^) exchange in essential metabolic processes and scavenging of intracellular ROS generated during the infectivity cycle. It is well established that metabolic stresses, including obesity, type 2 diabetes [36], smoking [37], heart failure [38], hypertension [39], nerve damage [40], and brain injury [41], deplete intracellular ATP and NAD+ in affected tissues with a dramatic induction of ROS and PARP-1 expression. Induced ROS levels also result in depleted NAD+ levels and an impaired antioxidant system, following inflammatory triggers (cytokine response and cellular activation/adhesion markers) that are hallmarks of aging, hypertension, diabetes, and obesity [42,43,44,45]. Despite this importance, the direct roles of these coenzymes in viral replication and antiviral defenses remain largely unexplored. Strikingly, PARP-1 is recognized as a master regulator of intracellular NAD+ and ATP pools. In this context, we showed that PARP-1-mediated PARylation of nuclear p53 maintains the intracellular ATP pool via transactivation of Proline oxidase (*POX*) mRNA [46]. With regards to diseases, studies have shown that increased age-associated lower NAD+ and increased ROS levels serve as strong predictors of SARS-CoV-2-associated in-hospital mortality [47,48]. A recent laboratory investigation revealed that non-canonical PARP isozyme (PARP-10) having an affinity for NAD+ was consistently upregulated by SARS-CoV infection [4]. Moreover, the abundant expression of PARP-1 across tissue types (Figure 3) implies functional significance in diseased conditions. *Based on these observations, we postulate that deficiency of NAD*+ *due to increased PARP activation may be a key factor related to the SARS-CoV-2 associated disease spectrum*. Oxidative stress induces PARP1, whose hyper activation depletes NAD+ and ATP levels, culminating in energy loss and subsequent cell death [49,50,51]. Overall, these processes enhance pro-inflammatory signaling. Hence, regulating the levels/activity of PARP-1 may serve as an important factor to maintain basal intracellular NAD+ levels and restore redox balance without inducing immune dysfunction. The supplementation of NAD+ or precursors is hypothesized to minimize disease severity in COVID-19 patients. *Nevertheless, the significance of this hypothesis needs experimental and clinical validation*.

Interferon signaling induces the PARP family of proteins (e.g., PARP-9, PARP-10, PARP-12, and PARP-14) causing inhibition of ADP-ribosylation, a mechanism that is presumed to impact viral translation [28,29]. In parallel, it should be noted that the conserved macro domain of SARS-CoV species suppresses host interferon and immune response to facilitate viral replication [28,29]. Regulation of PARP-1 expression and activity is shown to be important in cancer, cell metabolism, stress response, DNA damage, and neuronal differentiation and signaling, in addition to drug abuse [18,52,53,54]. Similarly, other studies have also established the regulation of PARP-1 in several disease conditions, including hypertension, obesity, and inflammation [50,55,56,57].

We have also reported the in vivo and in vitro implications of post-transcriptional mechanisms regulating PARP-1 in the context of neuronal function [52,53]. Employing molecular and biochemical approaches, we established the role of cellular microRNAs miR-124 and miR-125b in the regulation of PARP-1 mRNA in neurons [52,53]. miRNAs are small non-coding RNAs that post-transcriptionally regulate gene expression by binding to 3′-UTR of mRNAs. Although several pieces of evidence exist for the binding of miRNAs to 5′-UTR, and the open reading frame (ORF) of mRNAs to induce post-transcriptional events [57,58,59,60], miRNAs have been shown to be important in the regulation of PARP-1 in various physiological and pathological conditions (Table 1). As presented in Table 1, several of these diseases involving regulation of PARP-1 have been well documented as risk factors for COVID-19-associated mortality and hospitalizations, thus highlighting the plausible role of miRNA-regulated PARP-1 in COVID-19.

One of the long-term effects of COVID-19 is a broad spectrum of neurological disorders [72,73,74]. Cerebrovascular complications featuring peripheral nervous system damage (olfactory disorder), and ischemic and acute hemorrhagic encephalopathies with neurological features, are some of the profound observations made in COVID-19 patients [75,76,77]. The neurological features include ischemic stroke and cerebral hemorrhage, reduced consciousness, and nerve pain with abnormal levels of blood pressure. Additionally, multifocal lesions in the thalamus with signs of hemorrhage were observed in COVID-19 patients [75,78,79]. It is widely postulated that SARS-CoV-2 may employ the “Trojan horse model” of viral pathogenesis in the CNS [80,81]. The viral particles traverse through the blood brain barrier (BBB) via transcytosis and paracellular mechanisms, predominantly via the destabilization of endothelial tight junctions [82,83,84]. The hijacking of integral transport mechanisms by viral particles from the blood stream to the brain parenchyma is exacerbated as a result of induced matrix metalloproteinase (MMP) activity, causing severe BBB dysfunction and a productive infection cycle in the cellular constituents of brain parenchyma [83,84]. Given that PARP-1 is ubiquitously expressed across all tissue types with higher expression in the brain (Figure 3), it is plausible that PARP-1 may be exploited as a useful therapeutic target.

Specifically, regarding the neurological and related pathologies observed in COVID-19 patients, it is critical to understand whether:(a)SARS-CoV-2 infection affects PARP-1expression across cell types of the brain;(b)SARS-CoV-2 infection in the brain alters intracellular ATP and NAD+ levels;(c)Alteration occurs in the expression levels of miRNAs that are known to regulate PARP-1 expression in the brain and whether there is a cell-type specific effect.


*Answering these broadly significant questions may provide key insights into the role of PARP-1 in the replication features of SARS-CoV-2 in the brain.*


The regulation of PARP-1, particularly by miRNAs in the context of COVID-19 pathogenesis, may be of therapeutic relevance. miRNAs have been proposed as novel therapeutic molecules for several diseases, including cardiovascular dysfunction, cancer, and viral infections [85]. More recently, studies have reported alterations in host cell miRNAs in COVID-19 infection and have proposed computational modeling to predict miRNAs that can target the viral genome [86,87,88]. It will be interesting and worth validating to identify if any of the PARP-1 regulatory miRNAs are altered in COVID-19 and whether there is any tissue/cell-type specificity in such alterations. Such studies are anticipated to advance our understanding of the role of miRNAs, and to establish the relevance of miRNAs for RNA-based therapeutics for COVID-19. Given that levels of miRNA can regulate gene expression, miRNA-replacement therapy is postulated as one of the viable and emerging therapeutic options. Nonetheless, limitations such as tissue specific delivery strategies and therapeutic feasibility of miRNAs need to be thoroughly investigated. 

miRNA and small RNA-based drugs are anticipated to be the next generation drugs for the cure and prevention of complex human diseases [89,90,91]. In contrast, there are currently four approved small molecule inhibitors of PARP-1, namely, Olaparib, rucaparib, talazoparib, and niraparib, which target PARP-1′s function and activity at the protein level and are used for several forms of cancer [92]. However, these inhibitors are non-selective inhibitors of PARP-1 [92]. Additionally, challenges including off-target effects, tissue distribution, dose optimization, and toxicity have resulted in a low success rate of small molecule inhibitors in clinical trials [93,94]. Nonetheless, the pleiotropic nature of miRNAs presents a major challenge in its therapeutic approach. However, miRNAs can be engineered with chemical modifications to enhance stability and target specificity; details are provided in these cited studies [90,95,96]. Importantly, the small RNA-based therapeutic approach is highly specific, and its off-target effects can be controlled and can bypass the secondary effects of targeting other mRNAs or protein function [97,98]. Although most oligonucleotide/nucleic acid-based drugs are antisense oligos or siRNA-based, the miRNA-based approach has yet to be proved to be successful. Nevertheless, a well-validated repertoire of miRNAs that regulate PARP-1 must be investigated in pre-clinical models. Therefore, targeting PARP-1 at the mRNA level by modulating the expression of tissue-specific microRNAs can be a path forward to examine the relevance of this idea. In conclusion, gaining insight into the effect of SARS-CoV-2 on miRNAs that regulate PARP-1 expression in tissue types will be an important step to understand the relevance of the “miRNA-PARP-1” axis in COVID-19 disease pathology. 

## Figures and Tables

**Figure 1 ncrna-07-00060-f001:**
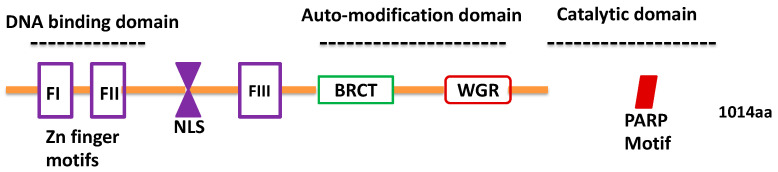
Structure of Poly (ADP)-ribose polymerase-1 (PARP-1). The *N* terminus of PARP-1 contains the DNA-binding domain (DBD) constituting two zinc-finger motifs and a bipartite nuclear localization signal (NLS). The auto-modification domain constitutes a breast cancer (BRCA) C-terminus-like (BRCT) interaction domain and the Tryptophan-Glycine-Arginine (WGR) domain that mediates interactions with self or protein partners. PARP-1 has a catalytic domain at its C’ terminus, within which is contained its Poly (ADP)-ribosylation polymerase (PARP) signature motif that catalyzes Poly (ADP)-ribosylation (i.e., PARylation) reactions using intracellular nicotinamide adenine dinucleotide (NAD+) as a donor of ADP-ribose.

**Figure 2 ncrna-07-00060-f002:**
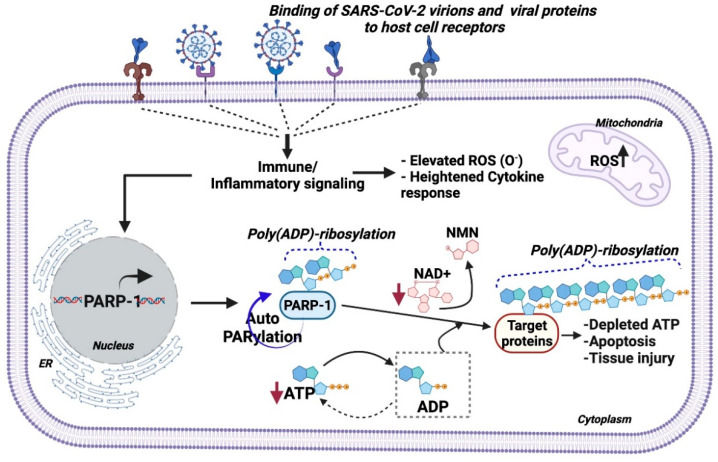
Model for PARP-1 activation in SARS-CoV-2 pathogenesis. Systemic inflammation in SARS-CoV-2 pathogenesis induces immune signaling in cells causing elevated mitochondrial ROS and cytokine response, which triggers PARP-1 transcription and activity. Thereafter, PARP-1 activation PARylates self (autoPARylation) or downstream protein targets by adding Poly (ADP)-ribose residues in a NAD+ dependent manner. Sustained PARylation events under diseased and inflammatory conditions cause declines in intracellular ATP and NAD+ pools, which leads to metabolic dysfunction and cell death, followed by tissue injury, as reported in COVID-19 pathology.

**Figure 3 ncrna-07-00060-f003:**
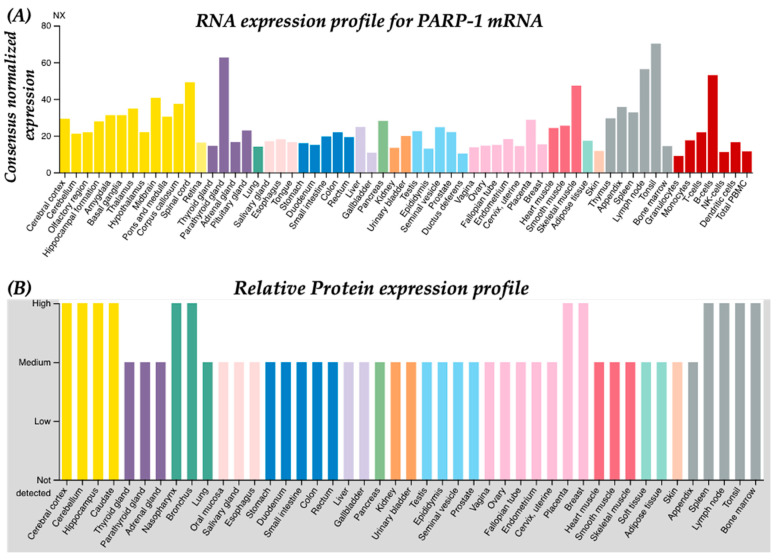
PARP-1 is expressed in all tissue types. (**A**) Normalized RNA expression profile for PARP-1 mRNA across human tissues. (**B**) Relative protein expression profile for PARP-1 across human tissues. These figures are obtained from the publicly available human protein and cell type atlas at proteinatlas.org. Details of data analysis, annotation, and normalization can be obtained at https://www.proteinatlas.org/ENSG00000143799-PARP1/tissue (accessed on 17 September 2021).

**Table 1 ncrna-07-00060-t001:** List of functionally validated cellular microRNAs that regulate PARP-1mRNA in the context of human diseases.

miRNA	Functional Assay	Region	Disease Relevance/Context	Ref.
miR-181a	Gof/Lof	3′-UTR	Acute Myeloid Leukemia	[61]
miR-7-5p	Gof/Lof; Reporter assay	3′-UTR	Small cell lung cancer	[62]
miR-379-5p	Luciferase Reporter assay	3′-UTR	Premature ovarian insufficiency	[63]
miR-103a-2-5 pmiR-585-5p	Luciferase Reporter assay	ORF	Oxidative stress, Hypertension	[64]
miR-335	Luciferase Reporter assay	3′-UTR	Small cell lung cancer	[65]
miR-520	Luciferase Reporter assay	3′-UTR	Recurrent spontaneous abortion	[66]
miR-223	Gof/Lof; Reporter assay	3′-UTR	Pulmonary arterial hypertension	[67]
miR-489	Luciferase Reporter assay	3′-UTR	Ischemic kidney injury	[68]
Let-7a	Luciferase Reporter assay	3′-UTR	HER2-overexpressing Breast cancer	[69]
miR-149	Luciferase Reporter assay	3′-UTR	Skeletal muscle metabolism	[16]
miR-577	Luciferase Reporter assay	3′-UTR	Myocardial infarction	[70]
miR-221-3p	Luciferase Reporter assay	3′-UTR	Triple negative Breast cancer	[61]
miR-124	Gof/Lof; Reporter assay, RNA-pulldown	3′-UTR	Drug (cocaine) abuse	[52]
miR-125b	Gof/Lof; Reporter assay, RNA-pull-down	3′-UTR	Drug (cocaine) abuse; Regulation of HIV integration	[53,71]

Abbreviations used: Gof/Lof, gain-of-function/loss-of-function; 3′-UTR, 3′-untranslated region; ORF, open reading frame.

## Data Availability

Not applicable.

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
