# Peer review of "Therapeutic Significance of microRNA-Mediated Regulation of PARP-1 in SARS-CoV-2 Infection"

_ncrna, 2021, doi:10.3390/ncrna7040060_

Round 1
Reviewer 1 Report
In this manuscript, the author Sabyasachi et al., introduced the general functions and structure of PARP-1, summarized the roles of PARP-1 in viral pathology, highlights the potential of PARP-1 as a druggable host gene for antagonizing viral infection especially SARS-CoV-2 and discuss the possibility of the usage of PARP-1 regulatory miRNA as a therapeutic agent. In general, the manuscript is well written, and the diagram is also very helpful. The authors provided important and insightful perspectives to this field. Here are my comments that may help the authors further improve this manuscript:
- Line 88. Coronaviruses are also RNA viruses. The tone sounds excluded. The authors may revise this to “although these findings are primarily reported for other RNA viruses”.
- Line 90-91. The authors should cite these studies.
- Inhibition of an anti-viral protein antagonizing the virus is surprising and thus the authors would better to provide some insightful discussion of its mechanisms.
- As the authors have mentioned that the PARP1 inhibitors are under clinical trials while they proposed to use miRNA to target this protein. The authors need to discuss the benefits of using miRNA over the inhibitors.
Author Response
Dear Reviewer,
Thank you for the valuable revisions and comments to improve our study. We have made the necessary updates and modifications addressing the comments and request you to kindly use track changes option to maintain the line numbers as explained below-
1) We have included the word "other", as suggested to clarify the tone and meaning of the sentence which is Line 93 in Page 3 of the revised version.
2) We have added the most recent citations as suggested for line 90-91, which is now line 96-97 in Page 3 of the revised version
3) In response to this comment- "Inhibition of an anti-viral protein antagonizing the virus is surprising and thus the authors would better to provide some insightful discussion of its mechanisms". We have clarified the ideas pertaining to this comment by integrating the following section in line 52-54 of Page 2 of the revised version (in bold): However, the exact mechanisms underlying the antiviral response of PARP superfamily of proteins are not clearly understood [17,18]. This is because several viruses harbor the machinery to modulate PARP function implying that PARP-mediated signaling can be both anti and pro-viral in a context dependent manner [18]. However, with regards to coronavirus species, in a recent study it was reported that SARS-CoV-2 infection strikingly induced activity of PARPs along with expression of genes encoding enzymes for salvage NAD synthesis from nicotinamide, and nicotinamide riboside, while simultaneously down-regulating other NAD utilizing biosynthetic pathways [4,19,10,13].
We also clarify that the effect of PARP-1 or PARP family of proteins can be both anti- and pro-viral depending on the virus type.
4) In response to the benefits of miRNAs over clinical inhibitors, we have added the following information (in bold) in Line 225-242 of Page 7 of the revised version: In contrast, currently there are four approved small molecule inhibitors of PARP-1 namely Olaparib, rucaparib, talazoparib and niraparib, that target PARP-1’s function and activity at protein level and are used for several forms of cancer [89]. However, these inhibitors are non-selective inhibitors of PARP-1 [89]. Additionally, challenges including off-target effects, tissue distribution, dose-optimization, and toxicity to name a few have resulted in a low success rate of small molecule inhibitors in clinical trials [90, 91]. Nonetheless, the pleiotropic nature of miRNAs presents a major challenge in its therapeutic approach. However, miRNAs can be engineered with chemical modifications to enhance stability and target specificity [92,93,94]. Importantly, the small RNA-based therapeutic approach is highly specific whose off-target effects can be controlled and can bypass the secondary effects of targeting other mRNAs or, protein function [97,98]. Even though most of the oligonucleotide/nucleic acid-based drugs are antisense oligos or siRNA, miRNA-based approach is yet to be successful. Nevertheless, a well validated repertoire of miRNAs that regulate PARP-1 must be investigated in pre-clinical models, Therefore, targeting PARP-1 at the mRNA level by modulating the expression of tissue specific microRNAs can be a path forward to examine the relevance of this idea.
Reviewer 2 Report
The authors present a position paper on the possible interplay between miRNAs and PARP(1). In general, I have no major issues with the paper, if the authors feel appropriate, they may add a discussion on PARP2 that in many respect phenocopies the behavior of PARP1 and is targeted by clinical PARP inhibitors (see PMID: 22581363, PMID: 18353725). PARP2 is also regulated by miRNA149 (PMID: 24757201).
Author Response
Dear Reviewer,
Thank you very much for the comments and providing us insights into the role of PARP2, whose function overlaps with PARP-1. The main idea in this perspective is to highlight PARP-1, whose expression is accounted in majority, and whose function is well studied/understood than other family members.
The abundance of PARP-2 is second to PARP-1 in the mammalian system, and recently its molecular role is gathering a lot of attention. Given this rising importance, we have included a section (in bold) in Line 45-47 of Page 1-2 to cite the importance of PARP-2 in the context of viral diseases.
Like PARP-1 function, recent studies have highlighted the role of PARP-2 in response to DNA damage, genome maintenance, metabolic regulation, oxidative stress [85,86,87].
Reviewer 3 Report
In this Perspective, Dash et al. describes the current literature pertaining to the SARS-CoV2 target, PARP and its regulation by non-coding RNAs. Specifically, the authors are interested in the therapeutic potential of miRNAs against PARP to prevent COVID-19 pathological outcomes.
- The authors speculate that NAD+ deficiency is due to decreased PARP activation, which is related to COVID infection/outcomes. A further explanation in the discussion on why targeting PARP via miRNAs has greater therapeutic potential/ is more cost-effective than supplementing with NAD+ for example, is required. Additionally, at what stage of infection should this pathway be targeted?.
- Is there any evidence that the COVID-19 vaccine produces a change in PARP response?
- Figure 3 should have titles clearly labeling graphs for mRNA and Protein. “A” and “B” should be vertically aligned. The grey box surrounding “B” should be removed. Labels need added for the Y axis on both A and B panels. Specifically, it isn’t clear what the numbers in A represent – percentages? If percentages, what reference tissue was used to determine 0 and 100%? What was the numerical cutoffs for determining “high”, “med” and “low” protein levels?
Author Response
Dear Reviewer,
Thank you very much for taking the time to provide us with valuable comments and suggestions to improve our study. We have made the necessary modifications to the best feasible way to improve the overall quality and value of the proposed ideas.
IN RESPONSE TO,
1) "The authors speculate that NAD+ deficiency is due to decreased PARP activation, which is related to COVID infection/outcomes. A further explanation in the discussion on why targeting PARP via miRNAs has greater therapeutic potential/ is more cost-effective than supplementing with NAD+ for example, is required. Additionally, at what stage of infection should this pathway be targeted?.
We apologize for any confusion with this idea. To clarify, NAD+ deficiency is not due to PARP-1's deficiency, instead the opposite. We have discussed that NAD+ deficiency is due to PARP-1's hyper activity or over activation since, PARP-1's activity is dependent on the availability of NAD+ as a cofactor. We have explained this concept through the model in Fig. 2 and Line 136-144 in Page 4 of the revised version. We have italicised this concept specifically in the Line 136-137.
2) This is an important question. However, as of today, we have not come across any report that state on the effects of approved therapies for COVID-19 on PARP-1 expression (global or tissue/cell type).
3) Thank you for pointing out regarding Figure 3: We have improved the quality of figure 3 as well as have labeled it clearly. We understand the color consistency with Figure 3B. It is important to know that these figures are adapted from Tissue expression atlas (https://www.proteinatlas.org/ENSG00000143799-PARP1/tissue). It is therefore not feasible to remove the background (grey) for figure 3B. We tried and tested various approaches to clear the background, however it significantly diminishes the quality and legibility of the figure. Hence, we chose to stick with the original.
Now, to add more clarity we have improvised the figure legends stating the details of how these figures were obtained. The idea behind presenting this figure is to highlight the abundant expression of PARP-1 across human tissues, which is an important factor for any candidate/molecule to qualify as a therapeutic target. The abundance of PARP-1 makes it easier to study its biology and design novel methods/ways for therapeutic targeting.